# Effectiveness of Somatosensory Stimulation for the Lower Limb and Foot to Improve Balance and Gait after Stroke: A Systematic Review

**DOI:** 10.3390/brainsci12081102

**Published:** 2022-08-19

**Authors:** Alison M. Aries, Poppy Downing, Julius Sim, Susan M. Hunter

**Affiliations:** 1School of Allied Health Professions, Faculty of Medicine and Health Sciences, Keele University, Keele ST5 5BG, UK; 2Royal Wolverhampton NHS Trust, Wolverhampton WV10 0QP, UK

**Keywords:** stroke, feedback, sensory, physical stimulation, sensory retraining, lower extremity

## Abstract

This systematic review’s purpose was to evaluate the effectiveness of lower-limb and foot somatosensory stimulation to improve balance and gait post-stroke. PRISMA reporting guidelines were followed. Included studies: randomized controlled trials (RCTs), published in English with ethical approval statement. Studies of conditions other than stroke, functional electrical stimulation, and interventions eliciting muscle contraction, were excluded. AgeLine, AMED, CINAHL PLUS, EMBASE, EMCARE MEDLINE, PEDro, PsycARTICLES, PsycINFO, SPORTDiscus, Web of Science and Cochrane central register of controlled trials were searched from 1 January 2002 to 31 March 2022. Two authors independently screened results, extracted data and assessed study quality using Cochrane Risk of Bias 2 tool; 16 RCTs (*n* = 638) were included. Four studies showed a medium or large standardized between-group effect size (Cohen’s *d*) in favor of somatosensory stimulation, in relation to: customized insoles (*d* = 0.527), taping (*d* = 0.687), and electrical stimulation (two studies: *d* = 0.690 and *d* = 1.984). Although limited by study quality and heterogeneity of interventions and outcomes, with only one study’s results statistically significant, several interventions showed potential for benefit, exceeding the minimally important difference for gait speed. Further research with larger trials is required. This unfunded systematic review was registered with PROSPERO (number CRD42022321199).

## 1. Introduction

Stroke is common and rising exponentially, with the incidence of stroke at 12.2 million in 2019 [1]; an estimated 150,000 of these were in the United Kingdom [2]. The overall result is 101 million cases worldwide [1]. Around 85% of stroke survivors have somatosensory loss [3], with the lower limb affected in approximately 50% [4,5]. This is important because sensory deficits affect the ability to produce voluntary movement [6,7,8] and undertake activities of daily living [9]. Specifically, impaired sensation in the foot and ankle can adversely affect balance and walking [10,11]. The importance of rigorously assessing somatosensation has been highlighted, because of the influence of somatosensory input on motor control and rehabilitation outcome post-stroke [12]. Rehabilitation strategies to address the impact of somatosensory impairment on motor activity and function have not been thoroughly investigated [13], and there is a lack of research related to sensory impairment of the feet post-stroke [14].

Proprioceptive training has been shown to precipitate changes in the supplementary motor area, pre-frontal cortex, and contralesional neural networks, with modifications noted on functional magnetic resonance imaging [15]. Some of the more recent advances in technology for stroke rehabilitation highlight the importance of afferent feedback, e.g., when using exoskeletons [16] or brain computer interface therapies [12].

Despite knowledge relating to the important influence of somatosensation on movement, rehabilitation interventions directed at reducing sensorimotor impairment are often not provided in clinical practice [17,18,19]. The value of passive sensory training (e.g., electrical stimulation) has been highlighted by a systematic review [19]; however, the authors acknowledged that little research had focused on active sensory training (education, localizing and discriminating sensations, sensory recognition, hardness discrimination and proprioceptive training). Studies included in the review were limited by small sample sizes, heterogeneity of participants, and unreliable outcome measures. Furthermore, the review combined findings for the upper and the lower limbs despite known neurophysiological differences relating to upper-limb and lower-limb control [20]. Consequently, it is reasonable to expect differences in response to the same sensory retraining intervention provided to the upper and lower limb. In view of the recognized importance of somatosensory information [6,7,8] and potential differences between upper- and lower-limb recovery post-stroke, with control of walking more automatic [21] and upper-limb movements dependent upon an intact corticospinal tract [22], systematic reviews focusing on the value of somatosensory stimulation to the lower limb post-stroke are required.

One systematic review of sensory retraining for the leg post-stroke does exist [23]; however, there are limitations to this review in terms of the interventions and study designs included. Undertaking a further systematic review of previous research limited to just somatosensory stimulation (not active movement) that includes a more robust study design—randomized controlled trials (RCTs)—will enable important design implications for future studies to be understood based upon preceding literature [24].

The aim of this systematic review was, therefore, to assimilate and analyze information relating to somatosensory stimulation post-stroke that specifically targets the lower limbs.

## 2. Methods

### 2.1. Searching for Literature

Following registration of the study protocol with PROSPERO, an international prospective register of systematic reviews (Number: CRD42022321199), a systematic review was undertaken. This has been reported using the Preferred Reporting Items for Systematic Reviews (PRISMA) recommendations [25] as a guide. PICO principles were followed [26], identifying: *population* (adult stroke survivors ≥18 years in any setting, i.e., acute or community settings); *intervention* (somatosensory intervention involving sensory stimulation (mechanical or tactile, thermal, electrical) for the purpose of sensory stimulation only, and proprioception of the contralesional lower limb and/or foot); *comparison* (standard care, routine/conventional therapy, or placebo/control); and *outcome* (valid and reliable outcomes related to gait and/or balance). The searches were developed with, and undertaken by, a health librarian. Study type was limited to RCTs published in English language, with a clear statement of appropriate ethics approval. Studies were excluded if they involved: participants with neurological conditions other than stroke; functional electrical stimulation and other interventions with the purpose of eliciting muscle contraction; sensory stimulation combined simultaneously with active or active assisted movement e.g., proprioceptive neuromuscular facilitation; acupuncture; transcranial magnetic stimulation; transcranial direct-current stimulation; visual or auditory stimulation; or feedback only, including visual biofeedback. Conference abstracts or other ‘grey’ literature, including unpublished studies and theses, were also excluded.

A search of literature from 1 January 2002 to 31 March 2022 was undertaken on 4 May 2022, enabling insight from the last two decades. If databases commenced after 1 January 2002, they were searched from database inception. A summary of the search is presented in Table 1; the full searches, which were adapted appropriately (e.g., use of MeSH terms and free text) for the different databases, are available in the Appendix A). A pilot search ensured that searches were sensitive rather than specific [27]. The following electronic bibliographic databases were searched: AgeLine, AMED, CINAHL PLUS, EMBASE, EMCARE MEDLINE, PEDro, PsycARTICLES, PsycINFO, SPORTDiscus and Web of Science. In addition, the Cochrane central register of controlled trials (CENTRAL) was also searched. Important health databases were, therefore, included.

Following the search, duplicates were identified and removed; this was undertaken manually to ensure accuracy.

### 2.2. Screening for Eligibility

Two authors (AMA and PD) then, independently, manually screened all the remaining reports for eligibility in accordance with the inclusion/exclusion criteria (Table 2), firstly for title, then abstract. No tools were used to assist with screening. Full texts were sought and screened if deemed to be potentially relevant; appropriate reports were also retrieved. Reasons for not including studies/reports in the systematic review are given in Appendix A. The reference lists of the included studies were also screened [27] (a further 28 full texts were read); studies meeting the inclusion criteria were included, and reasons for not including studies/reports in this stage are documented in Appendix A. In the case of disagreement, the fourth author (SMH) was available to discuss potential eligibility for inclusion. The PRISMA flow diagram [28], shown in Figure 1, summarizes the number of records identified for inclusion and exclusion at each stage of the process.

### 2.3. Quality Assessment

Two authors (AMA and PD) assessed the quality of all the included studies using the Cochrane Risk of Bias 2 (ROB2) assessment tool [29]. The tool enables evaluation of five different domains: (1) bias arising from the randomization process; (2) bias due to deviation from intended interventions; (3) bias due to missing data; (4) bias in measurement of the outcome; and (5) bias in the selection of the reported results, with an overall risk of bias also advocated. Any disagreements were discussed, and the fourth author (SMH) was available, if required. To assist with accurate evaluation of bias within the reports, all corresponding authors of included studies were contacted via email and offered the opportunity to provide any of the following information: trial protocol, statistical analysis plan, trial registry information, a thesis reporting the trial, regulatory documents or the research ethics application. If the reports did not specifically describe how missing data were handled e.g., whether an intention to treat analysis was undertaken, it was assumed it was not undertaken. The data from the ROB2 assessment were entered into the Robvis software [30], enabling a visual representation of the results.

### 2.4. Data Extraction

Two researchers (A.M.A and P.D) also extracted pertinent data independently (e.g., participant characteristics, sample size, study design, details of the content of what was delivered in both the intervention and comparison groups, outcome variables, and results) from the reports for all included studies. A bespoke data extraction form was used that had been developed and piloted by two of the authors (A.M.A and S.M.H). No tools were used for data extraction. The specific outcomes of interest were balance measures (e.g., postural sway), displacement of center of pressures (with eyes open and closed), Berg Balance Scale, Postural Assessment Scale for Stroke (PASS), and gait measures (e.g., timed 10 m walk test, mean velocity of gait, 6 min walk test, timed up-and-go and Functional Ambulation Categories). All data extracted were checked for accuracy by A.M.A and P.D. The information extracted was tabulated to aid synthesis and analysis.

### 2.5. Data Analysis

A narrative synthesis was undertaken, with studies grouped and discussed according to the somatosensory intervention delivered. Passive sensory stimulation techniques included manipulation of the surface beneath the feet to alter proprioceptive input, focal/segmental muscle vibration, taping and sensory electrical stimulation. Another category (sensory retraining) included manual handling techniques, e.g., specific sensorimotor foot stimulation; these techniques often involve a more active type of sensory retraining, with attention and learning being key components of the therapy. The decisions relating to the synthesis group were made in accordance with the detail given about the intervention and agreed by three authors (AMA, PD and SMH). To assist with synthesizing the information contained within the reports and identifying potential functional benefits of the interventions, unadjusted mean differences and 95% confidence intervals and standardized effect sizes (where possible) were calculated at the post-treatment timepoint for appropriate functional activities; e.g., balance or gait outcome measures. For two-group studies, standardized effect sizes are represented by Cohen’s *d,* whereby the difference is given in pooled standard deviation units [31]. Thresholds for small, medium and large effects are 0.20, 0.50 and 0.80, respectively. For studies with more than two groups, Cohen’s *f* statistic was used as a standardized effect size [31]. This statistic expresses departure of the group means from the grand mean in pooled standard deviation units; thresholds for small, medium and large effects are 0.10, 0.25 and 0.40, respectively. Immediate post-intervention scores were used; any follow-up measures were not analyzed. The post-intervention time point was considered by the authors to be the most relevant to inform clinical practice.

## 3. Results

### 3.1. Selection of Studies and Data Collection and Management

In total, 638 RCTs were returned by the searches; the numbers from the different databases and registers are documented in the PRISMA diagram (Figure 1). After removal of 201 duplicates, 437 reports were screened, first by title and then by abstract, by two researchers (AMA and PD) and 36 full texts were then screened for inclusion, of which 12 were appropriate to include. Reasons for exclusion are summarized in Figure 1, and further details reported in Appendix A. An additional four studies were identified from citation searching from the included studies; again, further details relating to why articles were excluded are presented in Appendix A. At each stage, the fourth author (SMH) was available to resolve any disagreements.

The process resulted in 16 studies included in the systematic review, with a total of 655 participants. Sample sizes of the studies ranged from 16 to 109. In the control groups, the number of participants randomized was 332, with results from 306 participants analyzed, and in the intervention groups 323 participants were randomized, with results from 303 participants analyzed. Details of all participants (age, time post-stroke, gender, side affected, type of stroke) and group allocation are available in Table 3.

### 3.2. Effects of Somatosensory Stimulation

Various balance and gait outcome measures were used across the studies. Where possible, a gait speed/velocity or balance outcome assessing postural sway or difference in weight distribution was selected for analyzing effectiveness. The outcome measures are presented in Table 4. The baseline and post-treatment values and between-group estimates of treatment effect, together with standardized effect sizes, are also presented in Table 4.

#### Somatosensory Interventions

Various types of somatosensory stimulation were delivered across the 16 studies, with seven of the studies (372 participants) involving electrical stimulation (with no muscle contraction) either via transcutaneous neuromuscular electrical stimulation (TENS) (six studies) [32,33,34,35,36,37] or interferential therapy (one study) [38]. Four of these studies included an outcome measure for gait (gait velocity (cm/s)) [32,37], 10 m walk test [38] and the timed up-and-go [35]). The other studies included balance measures (postural sway, with eyes closed) [34,36] and the Postural Assessment Scale for Stroke (PASS) [33]. All but one of the electrical stimulation studies showed some indication of effectiveness for electrical stimulation as a potential intervention post-stroke. The between-group standardized effect sizes for these studies ranged from *f* = 0.049 in relation to gait velocity [37] to *f* = 1.984 in relation to the PASS [36]. All of the studies in the electrical stimulation category had sample sizes greater than 34 (range 34–109).

Three studies [14,39,40] included some form of sensory retraining or manual therapy, including sensorimotor foot stimulation and ankle mobilizations. No standard deviations were reported for the Lynch et al. study [14], so it was not possible to calculate a confidence interval or effect size. Additionally, for this study the number of participants recruited and analyzed was smaller than the number specified in the power calculation. There were just eight participants who completed the intervention in the Goliwas et al. study [39] (receiving eight hours of sensorimotor foot stimulation over six weeks), with a mean difference of 1.60% (95% CI –15.88, 19.08; *d* = 0.084) in percentage weight distribution with eyes closed. Eight participants were included in the Kluding and Santos study [40], who only received a total of forty minutes of ankle mobilization treatment over four weeks, with peak weight bearing percentage difference measured in sit-to-stand, with a mean difference of 7.19% (95% CI –7.00, 21.38; *d* = 0.543) in favor of the control group. Neither of these studies [39,40] was powered to explore effectiveness.

Three studies [41,42,43] (total *n* = 45) involved manipulating the surface under the feet, altering mechanical alignment and proprioceptive input, via changes in surfaces [41] or wearing bespoke insoles [42,43]. Study samples ranged from 16 (just eight in the experimental group) [41] to 50 (25 in the experimental group) [43], with the third study [31] only including a total of 24 participants (12 in the experimental, insoles group). The mean difference ranged from 0.00 m/s (95% CI –0.18, 0.18; *d* = 0.000) [42] for mean velocity to 16.8 m (95% CI –1.95, 35.55; *d* = 0.527) [43] for the six-minute walk test.

The final three studies [44,45,46] explored either focal muscle vibration to tibialis anterior and peroneus longus on the paretic side [44] or to the plantar surface of both feet [45], or taping to tibialis anterior, the calf and the ankle joint [46]. The changes in gait in response to treatment for these studies reveal potential benefits from these interventions, but the confidence intervals indicate that none of the studies reached statistical significance. In the Paoloni et al. study [44], the mean difference of 0.07 m/s (95% CI –0.04, 0.18; *d* = 0.400), indicates potential for the segmental vibration to improve gait speed. 

**Table 3 brainsci-12-01102-t003:** Details of participants and group allocations in included studies.

Study	Study Design, Sample Size	Outcome Measurement	Study Group	No of Participants	Sex M/F	Side of Paresis L/R	Age Mean (SD) (Years)	Time post-Stroke; Mean (SD)	Type of Stroke: Infarct/ Haemorrhage	No. Finished Intervention
Bayouk et al. (2006) [41]	Matched pairs RCT, *n* = 16	Before and after 8-week program, no follow-up	Task-orientated training (2×/wk for 8 wks), 30 min each session—different surfaces proprioception feet/ankles and/or vision manipulated (8 h, total 16 h)	8	6/2	6/2	68.4 (7.1)	7.10 (12.50) yrs	Not stated	8
			Task-orientated training eyes open, hard surface (total 16 h.)	8	3/5	4/4	62.0 (4.6)	5.70 (6.90) yrs	Not stated	8
Cho et al. (2013) [34]	Randomized placebo-controlled trial, *n* = 42	Before and after intervention, with follow-up next day	Physical therapy for 30 min prior to TENS, single session for 1 h (total 90 min)	22	14/8	Not stated	55.2 (11.5)	15.00 (4.90) months	15/7	22
			Physical therapy, 30 min prior to placebo TENS (total 90 min)	20	13/7	Not stated	55.7 (8.6)	13.90 (5.10) months	14/6	20
Ferreira et al. (2018) [42]	RCT, *n* = 24	Before wearing insoles and after 3 months of insole use	Postural insoles influencing muscle proprioception (3 months of insole use, unclear how long they were worn each day)	12	11/1	6/6	59.2 (10.4)	3.90 (1.50) yrs	10/2	12
			Placebo insoles, no corrective elements.	12	5/3	6/2	60.3 (13.3)	3.30 (1.10) yrs	6/2	8
Goliwas et al. (2015) [39]	Single-blinded RCT, *n* = 27	On first and last day of stay in rehabilitation facility, no follow-up	Standard 5×/wk, 6-wk. rehabilitation program (30 min × 25 sessions, plus 15 min of sensorimotor foot stimulation (8.3 h, total 18.75 h)	13	5/3	2/6	62.3 (9.4)	4.40 (3.10) yrs	8/0	8
			Standard therapeutic rehabilitation program (45 min × 25 sessions, total 18.75 h.)	14	7/8	5/7	67.7 (9.2)	4.10 (2.80) yrs	12/0	12
In et al. (2021) [46]	Double-blinded RCT, *n* = 40	One day before and one day after sit-to-stand training, no follow-up	Sit-to-stand training, 30 min/day, 5x/wk, 6 wks + taping on tibialis anterior (total 15 h training + tape left in situ, changed every three days)	20	Not stated	10/10	56.2 (10.4)	7.05 (2.78) months	Not stated	20
			Just sit-to stand training, no taping (total 15 h)	20	Not stated	11/9	55.1 (9.9)	6.80 (2.50) months	Not stated	20
Jung et al. (2017) [36]	Double-blinded RCT, *n* = 41	Before and after 6-week training, no follow-up	30 min TENS to peroneal nerve 5×/wk for 6 wks (15 h) + Sit-to-stand (STS) training, 15 min/day, 5×/wk for 6 wks (15 h) + therapy, 1 h a day, 5×/wk., for 6 wks, (total 52.5 h)	20	11/9	10/10	56.2 (10.4)	6.05 (2.70) months	12/8	20
			Placebo TENS 30 min 5×/wk, 6 wks. (15 h) + STS training and therapy (total 52.5 h)	21	12/8	11/9	56.3 (10.2)	6.60 (2.50) months	11/9	20
Kluding and Santos (2008) [40]	Pilot RCT, *n* = 17	Before and after 4-week training, no follow-up	30 min, 2×/wk therapy for 4 wks functional training + contralesional ankle joint mobilizations (5 min) 2×/wk for 4 wks (40 min ankle mobilizations) (total 4.67 h.)	8	4/4	4/4	55.5 (10.8)	18.30 (11.8) months	Not stated	8
			2×/wk therapy (30 min) for 4 wks involving functional training (total 4 h)	9	5/3	7/1	56.1 (13.7)	24.60 (15.7) months	Not stated	8
Lynch et al. (2007) [14]	Pilot single-blind RCT, *n* = 21	Prior to treatment and on completion of treatment, with 2-week follow-up	Daily 1-h group session+30–60 min/day individual therapy (according to need) +10, 30 min sensory retraining sessions for 2 wks (5 h sensory) (total 20 h.)	10	7/3	5/5	61.0 (15.8)	48.70 (31.1) days	9/1	10
			Daily 1-h group session + 30–60 min/day individual therapy session + standing same time period (eyes closed) and 30 min of relaxation techniques (supine, eyes closed) (total 20 h)	11	9/2	3/8	62.0 (12.3)	47.80 (27.7) days	9/2	11
Ng and Hui-Chan (2009) [37]	Randomized, blinded placebo-controlled clinical trial (4 groups), *n* = 109	At baseline, after 2 and 4 weeks of treatment, follow-up 4 weeks after	TENS + exercise (5×/wk (60 min) for 4 wks) (20 h TENS and 20 h exercise, total 40 h.)	27	21/6	17/10	57.8 (7.3)	4.70 (2.80) yrs	11/16	26
			TENS (5×/wk. (60 min) for 4 wks) (total 20 h)	28	24/4	18/10	56.5 (8.2)	4.90 (3.90) yrs	13/15	25
			Placebo stimulation + Exercise (total 40 h)	25	20/5	13/12	56.9 (8.6)	4.70 (3.40) yrs	15/10	23
			Control (No treatment)	29	20/9	20/9	55.5 (8.0)	5.00 (3.00) yrs	16/13	27
Önal et al. (2022) [45]	RCT, *n* = 36	At baseline, and post intervention, no follow-up	Conventional physical therapy (CPT) (5×/wk for 4 wks - three 45 min sessions and two 60 min CPT sessions), plus local vibration therapy (LVT) (80 Hz) to plantar region (both feet for 15 min 3×/wk) (3 h. LVT and 17 h. CPT) (total 20 h.)	15	9/6	7/8	60(9)	12(3–24)	10/5	15
			CPT (5×/wk for 4 wks) (total 20 h)	15	11/4	10/5	59(9)	14 (6–39)	7/8	15
Paoloni et al. (2010) [44]	RCT, *n* = 44	Before and after training, no follow-up	50 min physical therapy session, (3 ×/wk for 4 wks + segmental muscle vibration 120 Hz (30 min each session) (Total 6 h vibration, 10 h physical therapy) (total 16 h)	22	19/3	11/11	59.5 (13.3)	1.90 (0.59) yrs	Not stated	22
			50 min physical therapy session, (3 ×/wk for 4 wks) (total 10 h)	22	20/2	10/12	62.6 (9.5)	1.86 (0.61) yrs	Not stated	22
Park et al. (2014) [32]	Single-blind RCT, *n* = 34	One week before and one week after intervention, no follow-up	30 min exercise with a physical therapist (5×/wk for 6 wks) + TENS (total 15 h TENS during 15 h exercise) (total 15 h)	17 (but characteristics for 15)	12/3	10/5	71.2 (3.46)	18.70 (2.46) months	Not stated	15
			30 min exercise with physical therapist + placebo TENS (total 15 h placebo TENS during 15 h exercise) (total 15 h.)	17 (but characteristics for 14)	8/6	8/7	71.1(3.82)	18.60 (1.70) months	Not stated	14
Suh et al. (2014) [38]	Single-blind RCT, *n* = 42	Immediately before and one week after intervention, no follow-up	30 min standard rehabilitation + electrical stimulation—60 min single session, interferential current (total 1 hr)	21	15/6	Not stated	54.4 (12.1)	15.05 (4.90) months	14/6	21
			30 min standard rehabilitation + sham electrical stimulation - one session, interferential current (total 1 h)	21	14/7	Not stated	53.9 (12.4)	13.90 (5.10) months	15/5	21
Wang et al. (2021) [43]	Single blind randomized clinical trial, *n* = 50	At baseline, and 4 weeks from baseline, no follow-up	Conventional gait training (40 min once a day 5×/wk for 4 wks) + customized insoles (worn for a minimum of 1 hr daily)	25	19/6	17/8	56.0 (range 49.5–66.5)	130.36 (64.87) days	13/12	25
			Conventional training (40 min once a day 5×/wk for 4 wks) (total 13.3 h)	25	18/7	18/7	60.0 (range 54.0–65.0)	123.08 (54.06) days	16/9	25
Yan and Hui-Chan (2009) [35]	Single blind stratified RCT, *n* = 62	Prior to treatment, weekly during 3-week treatment, follow-up 8 weeks post-stroke	Standard rehabilitation (OT and PT) each 60 min + transcutaneous electrical stimulation (5×/wk. for 3 wks.) (TES) (total 15 h)	21	9/10	11/8	68.4 (9.6)	9.20 (4.40) days	16/3	19
			Standard rehabilitation (OT and PT) each 60 min + Placebo stimulation (5×/wk for 3 wks) (total 15 h)	21	10/9	11/8	72.8 (7.4)	9.90 (2.60) days	16/3	19
			Standard rehabilitation (OT and PT) each lasting for 60 min (5×/wk. for 3 wks. (total 15 h)	20	9/9	11/7	70.4 (7.6)	8.70 (3.30) days	15/3	18
Yen et al. (2019) [33]	Prospective, assessor-blinded pilot RCT, *n* = 42	At baseline, at end of two-week intervention, with follow-up two weeks later	Standard rehabilitation (30 min 5×/wk. for 2 wks. +TENS (total 5 h)	14	7/6	Not stated	58.4 (13.5)	1.54 (0.78) days	7/6	13
			Standard rehabilitation + NMES † (total 5 h)	14	7/6	Not stated	61.6 (9.3)	1.38 (0.51) days	6/7	13
			Standard rehabilitation (30 min 5×/wk for 2 wks) (total 5 h)	14	9/5	Not stated	61.4 (12.6)	1.36 (0.50) days	6/8	14

CPT = conventional physical therapy, Hr = hour, LVT = local vibration therapy, Mins = minutes, NMES = neuromuscular electrical stimulation, OT = occupational therapy, PT = physical therapy, SD = standard deviation, STS = sit-to-stand, TENS/TES = transcutaneous electrical (nerve) stimulation, Yr = year, Wk = week. † Reported for completeness but not analyzed or discussed because the intervention delivered to this group involved muscle contraction.

**Table 4 brainsci-12-01102-t004:** Estimates of treatment effect and standardized effect sizes.

Study	Outcome Measure	Group (*n*)	Baseline; Mean (SD)	Post-Treatment; Mean (SD)	Mean Difference (95% Confidence Interval) *	Standardized Effect Size
**Manipulation of the surface beneath the feet to alter proprioceptive input**						
Bayouk et al. (2006) [29]	Ten-meter walk test (s)	1. Experimental—task-orientated training on different surfaces (8)	20.8 (8.3)	18.3 (6.5)	–1.4 (–11.95, 9.15) ^$ #^	*d* = 0.123 ^#^
		2. Control—task orientated training hard surface (8)	22.4 (13.8)	19.7 (12.3)		
Ferreira et al. (2018) [31]	Mean velocity (m/s)	1. Experimental—postural insoles (12)	0.57 (0.15)	0.57 (0.19)	0.00 (–0.18, 0.18)	*d* = 0.000
		2. Control—placebo insoles (8)	0.61 (0.30)	0.57 (0.19)		
Wang et al. (2021) [41]	Six-minute walk test (m)	1. Experimental—conventional gait training and customized insoles (25; 24 analyzed)	Data unavailable	64.68 (32.12)	16.8 (–1.95, 35.55) ^$^	*d* = 0.527
		2. Control—conventional gait training (25; 23 analyzed)	Data unavailable	47.88 (31.67)		
**Sensory retraining including sensorimotor foot stimulation and ankle mobilizations**						
Goliwas et al. (2015) [32]	Difference in weight distribution (eyes closed) (%)	1. Experimental—standard rehabilitation and sensorimotor foot stimulation (8)	26.9 (16.9)	18.1 (17.3)	1.60 (–15.88, 19.08) ^$^	*d* = 0.084
		2. Control—standard rehabilitation (12)	18.9 (20.9)	16.5 (18.8)		
Kluding and Santos (2008) [35]	Peak weight bearing difference in sit-to-stand (%)	1. Experimental—functional training and ankle joint mobilizations (8)	20.59 (11.67)	23.96 (13.04)	7.19 (–7.00, 21.38)	*d* = 0.543
		2. Control—functional training (9; 8 analyzed)	26.28 (14.67)	16.77 (13.42)		
Lynch et al. (2007) [13]	Ten-meter walk test (s) **	1. Experimental—group session and individual therapy plus sensory retraining (10)	35	23	2	—
		2. Control—group session and individual therapy and relaxation (11)	26	21		
**Focal muscle vibration**						
Paoloni et al. (2010) [38]	Gait speed (m/s)	1. Experimental—physical therapy and segmental muscle vibration (22)	0.44 (0.13)	0.53 (0.13)	0.07 (–0.04, 0.18) ^$^	*d* = 0.400
		2. Control—physical therapy (22)	0.44 (0.21)	0.46 (0.21)		
Önal et al. (2022) [37]	Ten-meter walk test (s)	1. Experimental—plantar vibration therapy (18; 15 analyzed)	27.83 (30.69)	20.15 (18.74)	3.38 (–8.15, 14.91) ^$^	*d* = 0.219
		2. Control—conventional physical therapy (18; 15 analyzed)	18.15 (11.07)	16.77 (11.15)		
**Taping**						
In et al. (2021) [33]	Ten-meter walk test (s)	1. Experimental—sit-to-stand training and taping (20)	25.74 (4.62)	20.11 (4.40)	–3.11 (–6.01, 0.21) ^$^	*d* = 0.687
		2. Control—sit-to-stand training (20)	25.01 (4.40)	23.22 (4.65)		
**Electrical stimulation (TENS or interferential therapy)**						
Cho et al. (2013) [30]	Postural sway (eyes closed), (cm)	1. Experimental—physical therapy and TENS (22)	89.79 (21.78)	69.05 (71.11)	–9.15 (–41.98, 23.68) ^$^	*d* = 0.178
		2. Control—physical therapy and placebo TENS (20)	85.31 (16.30)	78.20 (15.17)		
Jung et al. (2017) [34]	Postural sway (eyes closed), (cm)	1. Experimental—conventional therapy, sit-to-stand training and TENS (20)	104.1 (35.9)	77.6 (24.7)	–27.00 (–52.04, 1.96) ^$^	*d* = 0.690
		2. Control—conventional therapy, sit-to-stand training and placebo TENS (21; 20 analyzed)	117.7 (50.9)	104.6 (49.5)		
Ng and Hui-Chan (2009) [36]	Gait velocity (cm/s)	1. TENS (28; 25 analyzed)	57.7 (26.3)	60.9 (24.8)	0.00 (–13.83, 13.83) 1 vs. 4	*f* = 0.049
		2. TENS + exercise (27; 26 analyzed)	47.9 (26.8)	66.6 (32.5)	6.00 (–11.98, 23.98) 2 vs. 3^$^	
		3. Placebo stimulation + exercise (25; 23 analyzed)	50.7 (24.5)	60.6 (29.7)		
		4. Control (29; 27 analyzed)	58.9 (24.9)	60.9 (24.8)		
Park et al. (2014) [39]	Gait velocity (cm/s)	1. Experimental—physical therapy and TENS (17; 15 analyzed)	45.81 (15.22)	52.89 (17.43)	3.49 (–10.97, 17.95) ^$^	*d* = 0.183
		2. Placebo—physical therapy and placebo TENS (17; 14 analyzed)	46.85 (20.07)	49.40 (20.50)		
Suh et al. (2014) [40]	Ten-meter walk test (s)	1. Experimental—standard rehabilitation and interferential current (21)	44.75 (18.40)	37.74 (15.70)	–6.22 (–14.95, 2.51) ^$^	*d* = 0.446
		2. Placebo—standard rehab and sham stimulation (21)	45.93 (13.22)	43.96 (12.04)		
Yan and Hui-Chan (2009) [42]	Timed up-and-go (s)	1. Experimental—standard rehabilitation and TENS (21)	Data unavailable	30.0 (13.5)	–11.10 (–30.59, 8.39) 1 vs. 2 ^$^	*f* = 0.181
		2. Placebo—standard rehabilitation and placebo TENS (21)	Data unavailable	41.1 (27.9)	–25.40 (–56.54, 5.74) 1 vs. 3 ^$^	
		3. Control—standard rehabilitation (20)	Data unavailable	55.4 (47.1)		
Yen et al. (2019) [43]	Postural Assessment Scale for Stroke	1. TENS—standard rehabilitation and TENS (14; 13 analyzed)	3.77 (2.35)	31.38 (5.39)	7.46 (1.50, 13.42) 1 vs. 2 ^$^†	*f* = 1.984
		2. NMES - standard rehabilitation and NMES (14; 13 analyzed)	2.77 (1.01)	23.92 (8.91)	13.38 (7.61, 19.15) 1 vs. 3 ^$^†	
		3. Control—standard rehabilitation (14)	3.21 (1.12)	18.00 (8.65)		

*d* = Cohen’s *d* standardized effect size, *f* = Cohen’s *f* standardized effect size, NMES = neuromuscular electrical stimulation, SD = standard deviation, TENS = transcutaneous electrical (nerve) stimulation. * calculated as experimental group minus comparator group; ** data only presented graphically—values are estimated from the graph and no measure of variance was available; ^$^ indicates the direction of effect favored the somatosensory stimulation group; # a matched-pairs study, but data on the SD of differences was not available and the confidence interval and effect size are calculated without regard to the matching and are likely, therefore, to be conservative estimates; † *p* ≤ 0.05.

### 3.3. Quality Assessment

The quality assessment undertaken using the Cochrane ROB2 tool indicated some bias within all included studies, as shown in Figure 2. Most studies were assessed as low-risk relating to the randomization process (13/16) and adherence to intervention (13/16). As some studies had missing data, or the studies were not blinded, only 11 out of 16 studies scored low risk in these domains. Domain five was not very discriminating with only one study [45] being judged as low-risk relating to the selection of reported results. This was because this was the only study with a trial registry record containing this information; no responses were received from the study authors to whom personal communication (email) was sent requesting further information to verify the selection of outcomes planned a priori. Overall, bias for 50% of the included studies was judged as high, with all other studies judged as having ‘some concerns’ according to the Cochrane ROB2 tool.

## 4. Discussion

This systematic review involved a robust search of the literature with analysis and synthesis of information from 16 pertinent studies relating to the effects of somatosensory stimulation applied to the lower limb and foot to improve balance and gait post-stroke. If relevant gait outcomes were available, they were selected for analysis, with the timed 10 m walk test as the first choice for evaluation; this outcome measure is advocated by the Stroke Recovery and Rehabilitation Roundtable (SRRR) [47]. If suitable gait outcomes were unavailable, balance outcomes were selected with the aim of meeting the inclusion criteria of the systematic review. As the purpose of the review was to consider the effects of sensory interventions, if there was a choice of balance measure—e.g., postural sway with eyes open or closed—the eyes-closed measure was selected, since the ability to balance in this situation would be more dependent upon proprioceptive input, a key aspect of afferent input.

Heterogeneity across studies needs to be acknowledged, with age ranging from 55 [46] to 70.5 [35] years, and time post stroke ranging from just 1.5 days [33] to 6.4 years [41]. Additionally, diverse outcome measures were used, and a range of somatosensory interventions were studied. Intensity of delivery of the interventions also varied, with regimens ranging from just forty minutes (five minutes twice a week for four weeks) of somatosensory stimulation [40] to 20 h (one hour, five times a week for four weeks) [37]. Sample sizes ranged from 16 [46] to 109 [35] participants.

A meta-analysis of the between group effects was not undertaken, because calculation of a pooled effect was not considered meaningful, given that there were a number of different interventions (and for all but one of these interventions, no more than three studies were included in the review). Additionally, within each type of intervention, various outcomes were presented and analyzed.

A few other systematic reviews have explored sensory retraining post stroke [19,23,48] and more specifically electrical simulation [49,50]. In the systematic reviews undertaken by Schabrun and Hillier [19] and Serrada et al. [48], sensory retraining for the upper limb was included as well as for the lower limb, with 64% of the studies in the former and 50% of the studies in the latter pertaining to the upper limb. The systematic reviews that focused on electrical stimulation also included both upper-limb and lower-limb interventions [49,50]. One systematic review of sensory retraining for the leg post-stroke does exist [23]; however, there are limitations to this review. One important issue is that 50% of interventions within the included studies involved active movement (e.g., gait re-education, assisted movement, active movement, treadmill training, and virtual reality training). It is not possible to know whether it was the sensory training or the active movement that ‘significantly improved somatosensory function’ (p. 964). Additionally, many different study types were included in the review, and importantly, some relevant databases (AMED and Web of Science) were not searched. Another key difference between our review and these three reviews [19,23,48] is that our review excluded any interventions where active movement occurred as part of the sensory intervention (e.g., movement away from a noxious thermal stimulus). We also excluded randomized cross-over trials or trials without an appropriate control group, where it was not possible to assess the specific effects of the somatosensory intervention. Some similarities in the results across all the reviews exploring sensory retraining are observed, with all the findings demonstrating the effectiveness of passive sensory training (via electrical stimulation) without muscle contractions, but limited evidence of the effectiveness of other interventions. It is important to consider the difference between passive sensory stimulation and active sensory retraining. Schabrun and Hillier [19] discuss delivery of passive interventions, involving external stimulation to prime the nervous system and active sensory retraining driving perceptual change through specific exercises involving attention. Some of the interventions within our systematic review involved the application of passive external stimulation (TENS, interferential therapy, focal/segmental vibration and taping), whereas others involved more active involvement from the participants. For example, the sensory retraining in the Goliwas et al. study [39] (sensorimotor foot stimulation) necessitated the participants learning and improving selective movements and symmetrical weightbearing. It is likely that in order to successfully drive perceptual changes and learning, more intensive treatment is required over a longer period of time. It is interesting to discover that the results from this systematic review support this theory.

Although the mean differences from six of the studies reveal a potential benefit from the delivery of electrical stimulation, the confidence intervals indicate that the results of all but one of the studies [33] did not reach a 5% level of statistical significance. Furthermore, the results need to be interpreted with a degree of caution because there could be alternative reasons for the improvements seen. For example, in the Yen et al. (2019) study [33], the mean difference was 7.46 (95% CI 1.50, 13.42), *f* = 1.984, with a significant effect demonstrated (*p* < 0.001) in relation to the PASS, following just two weeks of TENS intervention to tibialis anterior and quadriceps muscles, compared to a control group given standard rehabilitation alone. However, the participants (only 13 in the intervention group) were an average of just 1.32 days post-stroke, so although the results are impressive, much of these changes could have been due to spontaneous recovery and heterogeneity within the sample, with stroke survivors having various lesion sites and sizes and, therefore, different rehabilitation potential. Response over the first few days post stroke can be very variable, due, in part, to the potential recovery of the penumbra area around the lesion [51]. Nevertheless, some of the studies, for example, Suh et al. [38] (*n* = 21) recruited stroke survivors more than 12 months post-stroke (mean (SD) 15.05 (4.9) months) so the cohort would have been expected to have a stable baseline. The mean difference for this study following just a single intervention of 60 min of interferential therapy, plus standard rehabilitation, compared to rehabilitation plus sham stimulation, measured by the ten-meter walk test (seconds) was –6.22 (95% CI –14.95, 2.51), *d* = 0.446, in favor of the intervention group. Although this result did not reach statistical significance, it potentially achieved a clinically significant change. The mean difference of –6.22 s equates to 0.62 m/s change; it is suggested that a change of just 0.16 m/s is the minimal clinical important difference for the ten-meter walk test [52]. Consequently, some participants may have changed their gait speed to better meet the requirements for community ambulation, which is suggested to be a speed of 0.8 m/s [53]. However, it is important to consider that an increased speed of gait does not indicate improvement in its quality. Further research needs to be undertaken to explore symmetry of gait after somatosensory interventions such as electrical stimulation.

None of the of the studies exploring sensory re-training or manual therapy were appropriately powered to explore effectiveness, and the intensity of the treatments delivered may have been insufficient to demonstrate clinical changes. Hands-on sensorimotor training to the feet has specific aims, including improving sensory perception and mobilizing the soft tissues and structures within the foot, with an objective of improving placement of the foot on the floor and, thereby, improving weightbearing [39]. These are different potential effects from those anticipated from the delivery of sensory TENS, where reduction of spasticity may be the main objective; this was the case for several of the studies included in this review [32,34,35,36,38].

Future trials should consider carefully which outcome measures might be best to include. For example, assessment of asymmetry in gait might be more appropriate than simple gait velocity to gain a better understanding of whether there is good contact, or not, of the hemiparetic foot (on the contralesional side) on the floor during stance phase of gait compared to the foot on the ipsilesional side.

Studies have been undertaken exploring sensorimotor stimulation for the upper limb (similar to the intervention delivered in the Goliwas et al. study [39]), using a treatment schedule called Mobilization and Tactile Stimulation (MTS), which involves intensive hands-on proprioceptive stimulation. Replicated single-system studies exploring MTS in both acute stroke (*n* = 6) [54] and chronic stroke (*n* = 8) [55] resulted in clinically significant improvements in the Action Research Arm Test (ARAT) and the Motricity Index for both studies. In these studies, the intensity of the MTS delivered was for up to one hour five times per week for six weeks (i.e., approximately 30 h of MTS intervention delivery). A latency of effect was noted within the studies, with some participants not responding until they received many hours of treatment. Indeed, in the Winter et al. study [55], improvements were not seen for the ARAT until between five and thirty days. In a follow-up dose-finding study [56], the most effective and feasible dose for delivering MTS that was recommended was a mean daily dose of between 37 and 66 min (delivered over a period of two weeks in the dose-finding study). In view of these findings, it is probable that the interventions delivered in the Goliwas et al. [39] and Kluding and Santos [40] studies were of an insufficient intensity to effect a change. The stroke recovery trial development framework produced consensus-based core recommendations from the second SRRR, suggesting it is important to consider aspects relating to the intensity of treatment when designing trials [57]; further research is clearly required.

A cutaneous proprioceptive stimulus (afferent input), such as that delivered by MTS, is proposed to increase excitability in the central nervous system, facilitating motor activity by decreasing pre-synaptic inhibition in response to the enhanced proprioceptive input, with changes seen particularly in people severely affected by stroke [58]. MTS is thought to reawaken the limb, preparing the sensorimotor system prior to retraining motor activity and function, which in turn facilitates plasticity in response to subsequent practice of tasks (e.g., through task-specific training) [59]. It is acknowledged that neurophysiological mechanisms for the upper limb may differ from those controlling lower-limb function; for example, there is greater automation involved in lower-limb control for balance and gait, with functional activity of the lower limbs less reliant on an intact corticospinal tract because the reticulospinal, rubrospinal and vestibulospinal tracts all contribute to the control of lower-limb movement [20]. Contrary to this, upper-limb activity and dexterity require corticospinal tract connectivity for good control [22]. However, it is anticipated that many of the principles relating to the delivery of MTS to the upper limb can be applied to the lower limb. Feasibility and acceptability of delivery of MTS to the lower limb has already been confirmed [60]. In view of these many aspects, it is not possible to draw conclusions about manual sensory retraining for the lower limb from this current systematic review, except to suggest that further research is necessary in this area.

The results of the three studies [41,42,43] that manipulated the surface under the feet, altering mechanical alignment and proprioceptive input via changes in surfaces, did not reach statistical significance; however, they may have resulted in a clinically important difference. The sample sizes were relatively small; further studies in this area of research are required.

Further research is also required to validate the findings from the studies exploring either vibration or taping, because conclusions should not be drawn from single studies with small sample sizes; however, both the In et al. study [46] and the Önal et al. study [45] were powered to explore effectiveness. The result from the Paoloni study [44] is above the small meaningful gait speed change (0.06 m/s) suggested by Perera et al. [61], and in the Önal et al. study [45] the mean difference following focal vibration was 0.34 m/s over ten meters, which is certainly well above the minimal clinically important difference. A similarly impressive increase in gait speed in the In et al. study [46] for the ten-meter walk test equates to a 0.31 m/s increase, which is also well above the minimal important clinical difference of 0.16 m/s [52].

Another point to note is that none of the studies included in this review were deemed at low risk of bias when analyzed using the ROB 2 tool. There were robust methods used for many of the studies, with low risk assessed for randomization procedures (13 studies), deviations from intended interventions (13 studies), bias due to missing data (11 studies) and blinded assessment reported in most (11) studies. The overall risk of bias was, however, severely influenced by the results in domain 5 (bias in selection of the reported results), which was not discriminatory, with all but one of the studies being judged as ‘some concerns’. This was due to the lack of access to study protocols or trial registration and ethical approval details for all but one study [45]; it was, therefore, not possible to ascertain whether or not the outcomes analyzed were those originally stated in the statistical analysis plans for the studies. Six of the studies were published over ten years ago, when study reporting mechanisms were not so advanced, so perhaps this is an expected finding. Nevertheless, four of the studies were published in the last five years and it was surprising to find that information relating to a priori decisions for analysis was only available for the one study. All corresponding authors of the included studies were contacted via email and given the opportunity to send copies of any documents that would enable the judgement call of this section of the ROB 2 tool to be reviewed; however, no information was received from any author except Önal et al. [45].

### Strengths and Limitations of Our Systematic Review

This systematic review has numerous strengths. A rigorous search was undertaken; only RCTs involving a somatosensory intervention, without concurrent active movement (muscle contraction) as part of the sensory intervention, were included. Studies undertaking a somatosensory intervention alongside, for example, task-specific training, e.g., task orientated training plus TENS, were, however, included providing there was also a true control (with no somatosensory intervention), enabling evaluation of the effects of the somatosensory intervention. Two independent researchers screened the titles, abstracts and full texts, extracted the data and undertook the quality assessment. A robust tool, the ROB 2, was used to assess the methodological quality of the studies. A third researcher advised in case of disagreement. Authors of the included studies were contacted and offered the opportunity to provide additional information. The reporting of this systematic review aligns with PRISMA guidelines [25].

The main limitations are that both language bias (only articles in English were included) [27] and publication bias [62] may have affected the findings of this systematic review; it is acknowledged that studies with positive results are more likely to be published in peer reviewed journals, particularly when written in English [27]. Additionally, the authors of the included studies, with one exception, did not respond with additional detail as requested, so it was not possible to ascertain whether planned outcome measurements and analyzes aligned with those reported. Additionally, in most cases, information was not available that would have allowed the mean differences calculated to be adjusted for baseline differences in the outcome measure; the unadjusted differences presented, and the standardized effect sizes calculated from them, should, therefore, be interpreted circumspectly. Additionally, heterogeneity of both the interventions and the outcome measures in the studies included in the review prevented statistical pooling of data in a meta-analysis.

## 5. Conclusions

Despite the heterogeneity across the studies, mean differences between the intervention and control groups indicate that there are potential benefits (often to the level of a minimal important clinical difference) of including sensory stimulation in stroke rehabilitation. The findings from this systematic review support those of previous systematic reviews exploring sensory stimulation or retraining post-stroke; further studies are required to explore sensory stimulation, particularly active sensory retraining programs with larger sample sizes, adequate treatment intensity and robust designs.

## Figures and Tables

**Figure 1 brainsci-12-01102-f001:**
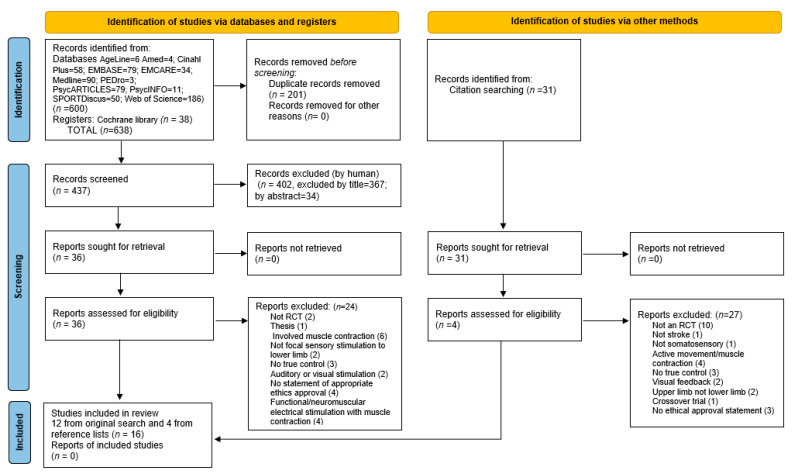
PRISMA diagram for systematic review [28].

**Figure 2 brainsci-12-01102-f002:**
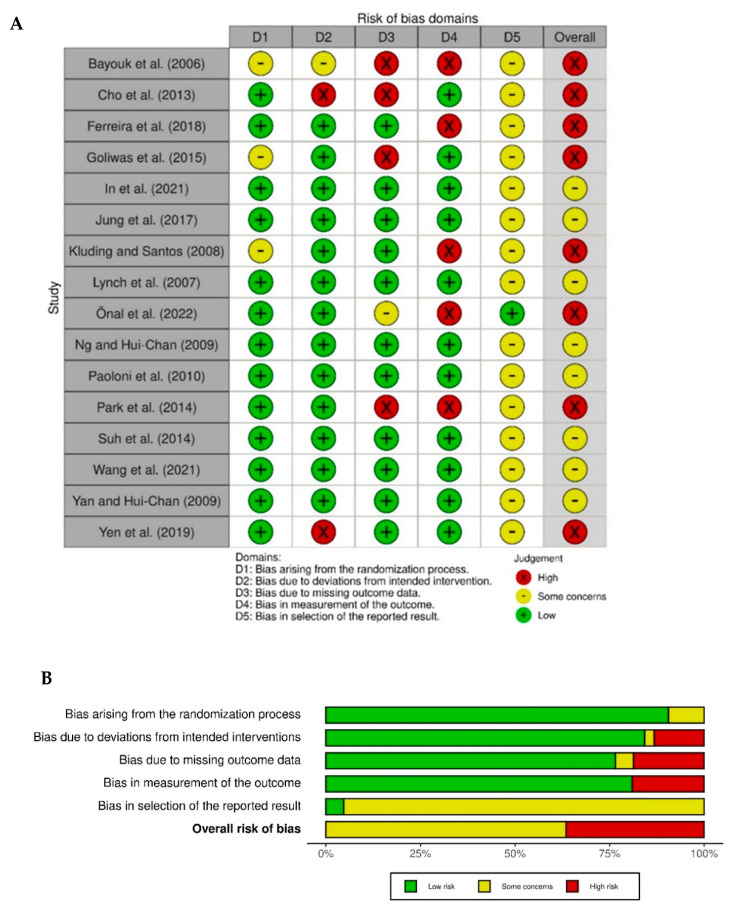
Risk of bias assessment using the Cochrane ROB 2 tool. (**A**) Review of authors’ judgements for each domain within the ROB 2 tool. (**B**) Risk of bias assessment summary—collective risk of bias of all studies (%) using the ROB 2 tool [14,30,32,33,34,35,36,37,38,39,40,41,42,43,44,45,46].

**Table 1 brainsci-12-01102-t001:** Search strategy for systematic review.

Aspect	Keywords and Boolean Operators
Population	“stroke” OR “cerebrovascular accident” OR CVA OR “acquired brain injury” OR “traumatic brain injury” OR “head injury” OR “TBI” OR “ABI” OR hemiplegia OR hemiparesis OR “upper motor neuron lesion”
AND	
Intervention	Sens* OR stimulat* OR somatosens* OR propriocept* OR afferent OR mobilisation OR mobilization OR manipulat*
AND	
Site	Foot OR leg OR “lower limb” OR “lower extremity”
AND	
Outcome of interest	Walk* OR gait OR mobil* OR step OR stance OR ambulat* OR “weight bear*
AND	
Type of study	Randomised controlled trial OR “randomised controlled trial” OR randomized controlled trial OR “randomized controlled trial”
NOT	
Main exclusion (to focus the literature search)	“Functional electrical stimulation” OR functional electrical stimulation OR FES

* has been used as a truncation symbol.

**Table 2 brainsci-12-01102-t002:** Inclusion and exclusion criteria for the systematic review.

Inclusion criteria
Adult stroke survivors aged ≥18 years
Somatosensory intervention involving sensory stimulation (mechanical or tactile, thermal, electrical for the purpose of sensory stimulation only, and proprioception) of the contralesional lower limb and/or foot
An appropriate control/placebo intervention
Gait and/or balance outcome measure
Randomized controlled trial
Published in English between 1 January 2002 and 31 March 2022
Appropriate ethical approval
**Exclusion criteria**
Any other condition, or inability to separate a cohort of stroke participants from other reported conditions
Active or active-assisted movement, as part of the specific sensory intervention; e.g., proprioceptive neuromuscular facilitation (if a separate intervention was delivered equally to all groups, such as conventional therapy or task-orientated training, in addition to specific sensory training in one group, the study was considered appropriate for inclusion)
Acupuncture
Transcranial magnetic stimulation or transcranial direct-current stimulation
Visual or auditory stimulation or feedback only, including visual biofeedback
Conference abstracts or other ‘grey’ literature, including unpublished studies and theses

## Data Availability

Data used in this review are available in the published studies included.

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
