# Peer review of "Effectiveness of Somatosensory Stimulation for the Lower Limb and Foot to Improve Balance and Gait after Stroke: A Systematic Review"

_brainsci, 2022, doi:10.3390/brainsci12081102_

Round 1
Reviewer 1 Report
This is a thorough and well- planned systematic review of the effectiveness of somatosensory stimulation after stroke . It is well-organized and written, but I have a few comments and questions.
Please explain in more detail how the publication bias might affect the reliability of your findings.
Although not necessary here as this review is well-organized, the inclusion of the completed PRISMA checklist for the reviewers would be a welcome addition.
Author Response
Please see attached file with our responses to the reviewer's comments

Reviewer 2 Report
Dear Authors,
thank you to giving me the opportunity to review your manuscript. This systematic review is focused on the effectiveness of somatosensory stimulation for lower limb in stroke survivor. The topic is very actual, because the research in new technologic in rehabilitation is a new challenge in scientific community. The somatosensory stimulation somatosensory stimulation approaches have been frequently used in order to help post-stroke patient to regain lost function, but its utility in the recovery of upper and lower limbs is still debated. The paper is well written and succint, but some critical issues should be addressed:
Introduction: he authors affirm that one systematic review is present in literature abuot somatosensory recovery in lower limb in stroke patient. This is ture, but it would be better to discuss this point in the discussion. I suggest to improve with more references the strong impact of somatosensory alteration in stroke patient, adding data about radiologic and physiolopathologic findings. Moreover, it should be desirable to stress the topic about the use of new technologie in rehabilitation of lower limb. Please, read "Calafiore D, Negrini F, Tottoli N, Ferraro F, Ozyemisci-Taskiran O, de Sire A. Efficacy of robotic exoskeleton for gait rehabilitation in patients with subacute stroke : a systematic review. Eur J Phys Rehabil Med. 2022 Feb;58(1):1-8. doi: 10.23736/S1973-9087.21.06846-5. Epub 2021 Jul 12. PMID: 34247470" and "Pillette L, Lotte F, N'Kaoua B, Joseph PA, Jeunet C, Glize B. Why we should systematically assess, control and report somatosensory impairments in BCI-based motor rehabilitation after stroke studies. Neuroimage Clin. 2020;28:102417. doi: 10.1016/j.nicl.2020.102417. Epub 2020 Sep 15. PMID: 33039972; PMCID: PMC7551360"
Methods:
Minor revision: Please, add different references every time to cite a study ( eg: two studies have-put references).
Major revision: Please, add notices about follow-up and the evaluation time for each study
Results: The results are not quite clear. The tables are very intuitive, but the written results are poor and should be add to make clearer the reading. Please, add paragraph to explain the different studies and results from each study. I suggest to define the paragrah with the different somatosensory tecniques.
Discussion: The discussion should be redefined. The authors mostly explain the results of the single studies and the real discussion start from line 434. Moreover, the authors add the PROSPERO number in the stength of the study, but this is incorrect. I suggest to rewrite it, focus the attention only on the discussion fo the single SS tecnique and the comparison between the present SR and the other in field
Best Regards
Author Response

(The authors gave the same response as above.)

Reviewer 3 Report
Thank you for submitting your manuscript to this journal. I read your manuscript with interest.
I am very happy to be able to write in a positive way but It may be that you would like to consider improving the manuscript in which case I hope that the comments from my review may help you to revise it before resubmitting it.
- Abstract: Regarding the keywords, please use MeSH-terms.
- Method, Discussion is well described.
- Refference: Please check the reference form.
journal name. year
ex) refference 26
BMJ: British Medical Journal. 2011, 343, 1-9.
Author name
ex) refference 30
Cho, H.Y.; In, T.S.; Cho, K.H.; Song, C.H.
Author Response

(The authors gave the same response as above.)

Reviewer 4 Report
Dear authors.
First of all, I greatly appreciate your work, however I have had some doubts that I hope you can solve for me.
Registration in POSPERO is never a limitation, this must be stated and explained in the methods section.
Second, the table with the inclusion and exclusion criteria is not clear enough, it would be better to reorganize it so that readers can understand it better.
I find your systematic review very interesting but I think it would contribute much more and would differ from other systematic reviews if you did a statistical analysis using the meta-analysis technique, why haven't you done this process?
In the table of the characteristics of the studies, it would be advisable to include the value of p, that would give your work a lot of scientific rigor.
Why does not appear a section where the results of the systematic review are shown? this is totally necessary for the reader
the results are in the discussion, that does not go there. You have to change it and improve it.
The discussion is practically non-existent, it must be improved and redone so that your publication can be valued.
Author Response

(The authors gave the same response as above.)

Round 2
Reviewer 2 Report
Dear authors,
thank you for your outstandig revision process. At the light of my knowledge, the paper is suitable for publication in journal.
Best Regards
Reviewer 4 Report
The authors have done a good job of revising and improving this paper.